# Interpretable Additive Tabular Transformer Networks

**Anton Frederik Thielmann**                    *anton.thielmann@tu-clausthal.de*
*Institute of Mathematics*
*Clausthal University of Technology*

**Arik Reuter**                    *arik_reuter@gmx.de*
*Institute of Mathematics*
*Clausthal University of Technology*

**Thomas Kneib**                    *tkneib@uni-goettingen.de*
*Chair of Statistics and Campus Institute Data Science*
*Georg-August-Universität Göttingen*

**David Rügamer**                    *david@stat.uni-muenchen.de*
*Department of Statistics, LMU Munich*
*Munich Center for Machine Learning (MCML)*

**Benjamin Säfken**                    *benjamin.saefken@tu-clausthal.de*
*Institute of Mathematics*
*Clausthal University of Technology*

**Reviewed on OpenReview:** *https://openreview.net/forum?id=TdJ7lpzAkD*

## Abstract

Attention based Transformer networks have not only revolutionized Natural Language Processing but have also achieved state-of-the-art results for tabular data modeling. The attention mechanism, in particular, has proven to be highly effective in accurately modeling categorical variables. Although deep learning models recently outperform tree-based models, they often lack a complete comprehension of the individual impact of features because of their opaque nature. In contrast, additive neural network structures have proven to be both predictive and interpretable. Within the context of explainable deep learning, we propose Neural Additive Tabular Transformer Networks (NATT), a modeling framework that combines the intelligibility of additive neural networks with the predictive power of Transformer models. NATT offers inherent intelligibility while achieving similar performance to complex deep learning models. To validate its efficacy, we conduct experiments on multiple datasets and find that NATT performs on par with state-of-the-art methods on tabular data and surpasses other interpretable approaches.

## 1 Introduction

Deep neural networks (DNNs) have emerged as one of the most powerful and versatile tools in AI, with remarkable abilities in modeling complex, high-dimensional problems and recognizing intricate patterns in non-tabular data. They have shown exceptional performances on tasks such as image classification (Yu et al., 2022; Dosovitskiy et al., 2020), text classification (Huang et al., 2021; Lin et al., 2021), audio classification (Nagrani et al., 2021), time-series forecasting (Zhou et al., 2022; Zeng et al., 2022) and many more. However, for the most common data type in real world applications, tabular data, DNNs were still outperformed by tree based methods as XGBoost (Chen et al., 2015) or LightGBM (Ke et al., 2017). In recent years, however, the transformer architecture (Vaswani et al., 2017) allowed for an increase in performances of tabular deep learning methods. Huang et al. (2020), Gorishniy et al. (2021) and Arik and Pfister (2021)

introduced architectures that were on par with state-of-the-art tree based models and outperformed baseline Multi-layer perceptrons. Gorishniy et al. (2022) introduced a model that even outperformed gradient boosting methods on a majority of popular tabular benchmark datasets. In addition to their increased predictive performance, attention based models for tabular data are semi-interpretable, by providing further insights through individual feature importances obtained from the attention layer(s). While main feature effects cannot be clearly identified or visually interpreted, the single feature importance can be abstracted from the attention layers in the transformer, allowing for a pseudo-feature significance.

While these models are incredibly powerful in predictive terms, true interpretability is still lost in their *black-box* nature. This ultimately limits their applicability as especially tabular data needs fully interpretable model structures to be applied in domains such as health care, finance or insurance. Explainability in these *black-box* models is often achieved with post-hoc analyses on the sample level with existing methods resorting to model-agnostic methods. Locally Interpretable Model Explanations (LIME) (Ribeiro et al., 2016), Shapley values (Shapley, 1953) or layer wise relevance propagation (LRP) (Bach et al., 2015) and their extensions (Sundararajan and Najmi, 2020) try to explain model predictions via local approximation and feature importance. Sensitivity-based approaches (Horel and Giesecke, 2020), exploiting significance statistics, can only be applied to single-layer feed-forward neural networks and can hence not be used to model complex non-linear effects. Although those approaches might be able to indicate how individual predictions are generated, they do not provide a global and complete picture of the underlying decision making process.

Recently, neural networks are making a push towards feature level interpretability by adopting the additive model structures from Generalized Additive Models (GAMs) (Hastie, 2017). GAMs use basis functions that transform features into higher dimensional space that allows feature effects to capture non-linearity. Neural Additive Models (NAMs) (Agarwal et al., 2021) proposed the neural counterpart to GAMs and inspired multiple adaptations which will be introduced in Section 2. One downside of these adaptions is that they treat all features identically, independent of the underlying data type. Numerical features are modeled the same way as categorical features, which leads to parameter-dense networks that can loose their easy interpretability. We introduce **N**eural **A**dditive **T**abular **T**ransformer (NATT) Networks, a model class that leverages the additivity constraint from GAMs and NAMs and implements transformer based embeddings for increased predictive performance while also allowing for an interpretable way to model categorical features.

In summary, NATT aims to narrow the performance gap between models that are inherently interpretable (e.g., GAMs (Hastie, 2017), NAMs (Agarwal et al., 2021), EBMs (Nori et al., 2019), NodeGAMs (Chang et al., 2021)) and black-box models. Additionally, NATT presents a strategy for enhancing the interpretability of networks handling categorical features. The issues with commonly used categorical encodings are discussed in detail in section 3.1. Crucially, when dealing with numerous categorical variables and particularly a wide array of categorical expressions, NATT offers scalability and maintains ease of interpretability, an area where traditional encodings implemented in models like NAMs struggle. Our proposed approach enables the interpretation of complex interactions among categorical expressions while maintaining a minimal number of shape functions, addressing the inherent challenges in scalable and interpretable modeling of categorical data. The contributions of the paper are as follows:

- We propose a novel model structure that incorporates categorical feature embeddings into an additive model architecture. This allows for the joint learning of all feature embeddings and shape functions.

- We demonstrate that NATTs outperform interpretable baselines as well as state-of-the art neural and tree-based boosting methods without loss of intelligibility.

- Our experimental results demonstrate that NATTs outperform their baseline counterpart, the Neural Additive Model (NAM), by an average of ~5% across multiple datasets and are on par with state-of-the-art *black-box* models when accounting for feature interactions.

- We demonstrate that NATT seamlessly incorporates pairwise or higher order feature interactions and find that $NA^2TT$ achieves superior performance compared to the interpretable baselines.

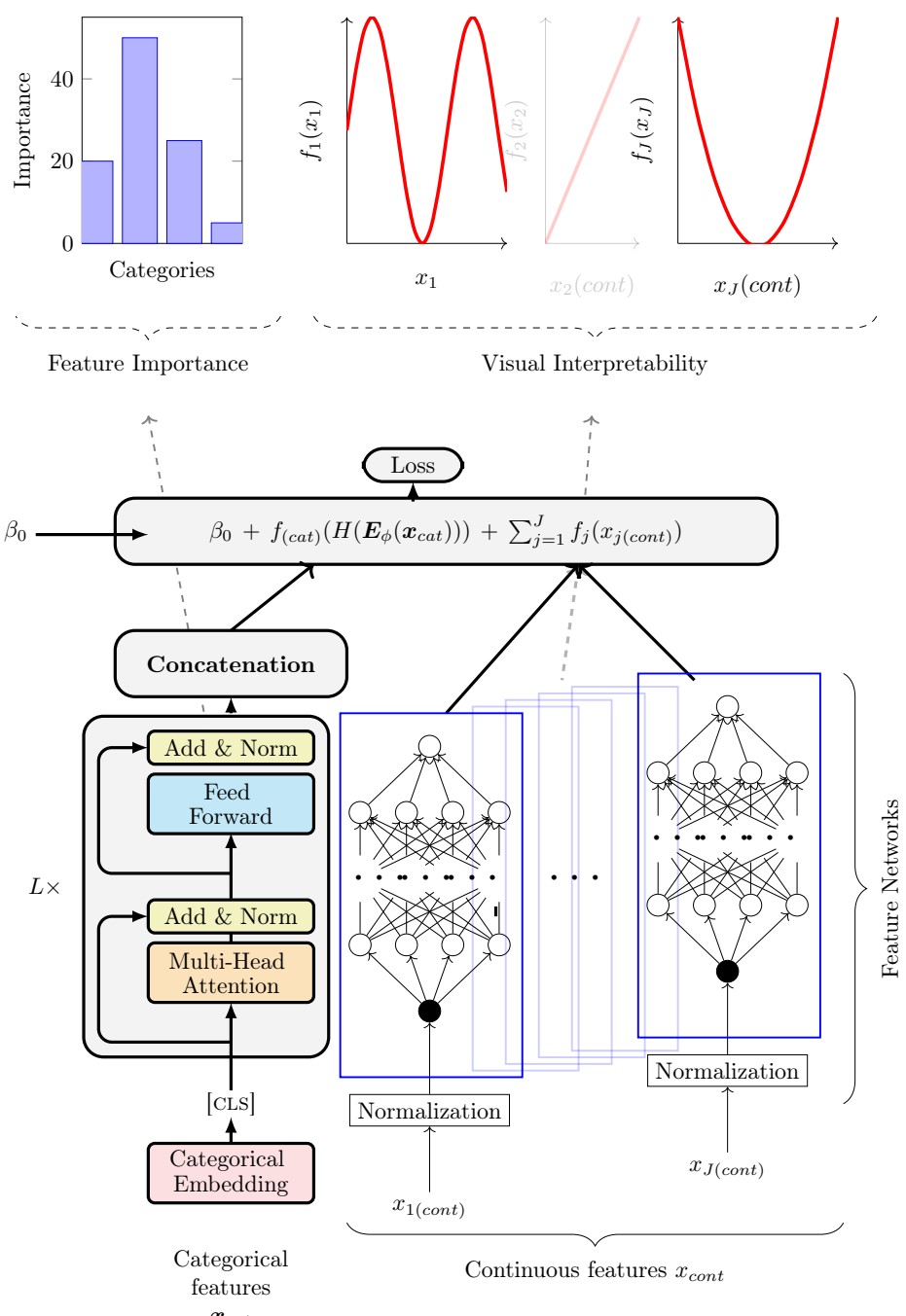

Figure 1: NATT model architecture and visually interpretable output. The continuous feature are visually interpretable thorugh the plotting the output of the independt feature networks. The categorical feature importance can be analyzed via the [CLS] token in the Transformer architecture.

## 2 Related work

The interpretability of deep neural networks has been a long-standing challenge in the field of Artificial Intelligence. While these models have achieved impressive accuracy in various tasks, their lack of transparency has limited their real-world applicability in some areas. This issue has led researchers to explore different approaches to increase the interpretability of these models, including generating feature-level interpretability.

One promising approach in this direction is the translation of Generalized Additive Models (GAMs) into a neural framework. This idea was first introduced by Potts (1999) and expanded by de Waal and du Toit (2007) in the late 90s and early 2000s. However, the previous approaches didn't use backpropagation and had limitations in their predictive power and interpretability.

In recent years, researchers have made significant progress in this field by introducing more flexible and effective approaches. One of these approaches is Neural Additive Models (NAMs), which were first introduced by Agarwal et al. (2021). NAMs use Deep Neural Networks (DNNs) instead of smooth shape functions from GAMs. This approach has led to several extensions, including adaptations of the additive model structure using different types of shape functions such as gradient boosted trees (Chang et al., 2021) or piecewise polynomials (Dubey et al., 2022).

Several approaches have been proposed to account for feature interactions (Kim et al., 2022; Tsang et al., 2018; Enouen and Liu, 2022). Wang et al. (2021) introduced Pie-GAMs, which are NAMs that incorporate a fully connected *black-box* Multi-Layer Perceptron (MLP) trained only on the residuals of a first NAM model not including feature interactions.

Luber et al. (2023) introduced extremely parameter sparse networks, leveraging basis expansion methods. Radenovic et al. (2022) switched the basis expansions for neural basis expansions, keeping the overall model structure from Agarwal et al. (2021) intact.

Additionally, researchers have considered the problem of distributional regression, which is essential in many real-world scenarios. Rügamer et al. (2023; 2021) introduced semi-structured distributional regression, combining the advances of deep neural networks with structured regressions. Thielmann et al. (2024) introduced Neural Additive Models for Location Scale and Shape (NAMLSS), the extension of NAMs for distributional regression.

## 3 Methodology

Let $\mathcal{D} = \{(\boldsymbol{x}^{(i)}, y^{(i)})\}_{i=1}^{n}$ be the training dataset of size n and $y$ denote the target variable that can be arbitrarily distributed. Each input $\boldsymbol{x} = (x_1, x_2, \ldots, x_J)$ contains J features. Given a link function $g(\cdot)$, that connects the linear predictor to the expected value of the response variable, accommodating different types of data distributions, a GAM in its fundamental form can be expressed as:

$$g(\mathbb{E}\,[y]) = \beta_0 + \sum_{j=1}^{J} f_j(x_j), \tag{1}$$

where $\beta_0$ denotes the global intercept or bias term and $f_j : \mathbb{R} \to \mathbb{R}$ denote the univariate shape functions corresponding to input feature $x_j$ and capturing the feature main effects. This model structure allows for easy interpretation of the feature effects as the shape functions can be visualized. In classical GAMs, the shape functions $f_j$ are smooth functions, e.g. polynomial splines. However, while being parameter sparse and interpretable, classical GAM smoothers lack the predictive power of current deep neural networks. Switching splines for e.g. Multi Layer Perceptrons (Agarwal et al., 2021) or decision trees (Chang et al., 2021) thus has shown to create extremely powerful yet interpretable models.

Independent of the type of shape functions, pairwise feature interactions can be integrated:

$$g(\mathbb{E}\,[y]) = \beta_0 + \sum_{j=1}^{J} f_j(x_j) + \sum_{j,k:j \neq k} f_{jk}(x_j, x_k), \tag{2}$$

where $f_{jk} : \mathbb{R}^2 \to \mathbb{R}$ denote the feature interactions between input features $x_j$ and $x_k$. Using neural networks as shape functions not only allows to model pairwise feature interactions (GA$^2$Ms), but higher order features interactions (GA$^J$Ms). These interactions can e.g. be modeled using a fully connected MLP (Kim et al., 2022; Wang et al., 2021). Depending on the shape function and the corresponding optimization method, regularization techniques, e.g. feature dropout as introduced by Agarwal et al. (2021) are used to account for identifiability.

Independent of the type of shape functions, however, all of these models have in common that they fit one shape function per feature main effect for all continuous or binary features. Categorical features, however, are either a) one-hot encoded – leading to each category being represented by an independent shape function (see, e.g., Agarwal et al., 2021; Radenovic et al., 2022; Enouen and Liu, 2022; Chang et al., 2021), b) target encoded (Chang et al., 2021; Popov et al., 2019) – such that dependencies between different categorical features cannot be captured and information may be lost if there are different categories with the same mean value of the target variable (Zeng et al., 2022), or c) label encoded (Nori et al., 2019) – which is problematic when the categories lack a natural ordering. Datasets containing a lot of categorical variables, or variables with a lot of categories can thus lead to increased model sizes and hardly interpretable feature effects due to the excessive amount of shape functions.

Arik and Pfister (2021), Huang et al. (2020), Hollmann et al. (2022) and most recently Gorishniy et al. (2022) demonstrated the advantages that transformer based embeddings can have for modeling categorical features. Let $\boldsymbol{x} \equiv (\boldsymbol{x}_{cat}, \boldsymbol{x}_{cont})$ denote the categorical and numerical (continuous) features, respectively with $\boldsymbol{x}_{cont} \in \mathbb{R}^c$. Further, let $x_{j(cat)}^{(i)}$ denote the $j$-th categorical feature of the $i$-th observation. In order to preserve interpretability and allow for complex feature interactions, we adapt (1) to:

$$g(\mathbb{E}\,[y]) = \beta_0 + \sum_{j=1}^{J} f_j(x_{j(cont)}) + f_{(cat)}(H(\boldsymbol{E}_\phi(\boldsymbol{x}_{cat}))), \qquad (3)$$

where $H(\cdot)$ denotes a sequence of transformer layers, creating the feature embeddings for the categorical features. The parametric embeddings $\boldsymbol{E}_\phi(\boldsymbol{x}_{cat})$ are the input of the first transformer layer. $H(\cdot)$ returns the contextualized embeddings $\{\boldsymbol{h}_1, \boldsymbol{h}_2, \ldots, \boldsymbol{h}_j\}$ that are subsequently fed into the shape function, $f_{(cat)} : \mathbb{R}^{(e \times J)} \to \mathbb{R}$, where $e$ denotes the chosen embedding dimension. The contextual embeddings, $\{\boldsymbol{h}_1, \boldsymbol{h}_2, \ldots, \boldsymbol{h}_j\}$, are thus jointly learned with the shape function, $f_{(cat)}$.

To embed each categorical feature into a *column embedding* we follow Huang et al. (2020). Our architecture for the categorical features comprises a column embedding layer, a stack of $L$ Transformer layers and a Multilayer Perceptron. The Transformer layers follow the classical architecture (Vaswani et al., 2017) and are comprised of multi-head self-attention layers and position-wise feed-forward layers. An embedding lookup table $\boldsymbol{e}_{\phi_j}(\cdot)$ is created for each categorical feature, $x_{j(cat)}$. For the j-th categorical feature with $d_j$ classes, the embedding table $\boldsymbol{e}_{\phi_j}(\cdot)$ has $(d_j + 1)$ embeddings with the additional dimension accounting for missing values. The set of embeddings for all categorical features is denoted by $\boldsymbol{E}_\phi(\boldsymbol{x}_{cat}) = \{\boldsymbol{e}_{\phi_1}(x_1), \boldsymbol{e}_{\phi_2}(x_2), \ldots, \boldsymbol{e}_{\phi_j}(x_j)\}$. Thus, the embedding for the encoded value $x_{j(cat)} = k \in [0, 1, 2, \ldots, d_j]$ is equal to $\boldsymbol{e}_{\phi_j}(x_{j(cat)}) = [\boldsymbol{c}_{\phi_j}, \boldsymbol{w}_{\phi_{kj}}]$. $\boldsymbol{c}_{\phi_j} \in \mathbb{R}^l$ denotes a unique identifier that distinguishes the classes from different columns. $\boldsymbol{w}_{\phi_{kj}} \in \mathbb{R}^{d_j - \alpha}$ and the dimension, $\alpha$, is a tuneable hyper-parameter just like the number of neurons in fully connected dense layers. To account for interpretability of the categorical variables, multiple adaptations are possible. Appending a [CLS] token to the column embedding and feeding the [CLS] into the shape function $f_{(cat)}$ similarly to Gorishniy et al. (2021), or leveraging sequential attention as done by Arik and Pfister (2021) could ensure interpretability and leave the overall model structure intact, without loss of generalizability or performance.

Note, that depending on the data structures, one could learn different feature embeddings and fit separate shape functions for different categorical features. However, as visually interpreting categorical features is not sufficiently meaningful, fitting one shape function for all categorical variables suffices. Additionally, jointly learning a single embedding vector for all categorical features, captures all possible interaction effects between all categorical variables while maintaining interpretable. Adapting for pairwise feature interactions between

the continuous features as well as the continuous and categorical features represented in (2) would lead to:

$$g(\mathbb{E}\left[y\right]) = \beta_0 + \sum_{j=1}^{J} f_j(x_{j(cont)}) + f_{(cat)}(H(\boldsymbol{E}_\phi(\boldsymbol{x}_{cat}))) + \sum_{j,k:j\neq k} f_{jk}(x_{j(cont)}, x_{k(cont)})$$
$$+ \sum_{j=1}^{J} f_{j(cat)}(x_{j(cont)}, (H(\boldsymbol{E}_\phi(\boldsymbol{x}_{cat})))), \tag{4}$$

where $f_{j(cat)}$ accounts for the feature interaction between continuous feature $j$ and all categorical features. Note, that the input dimension of $f_{j(cat)}$ is increased by 1, as the transformer output and the continuous feature are concatenated to form the input vector of the shape function.

### 3.1 Interpretability

One of the key advantages of GAMs is that the learned shape functions can be easily visualized. NATT inherits this feature from GAMs and the shape functions for all numerical variables are interpretable (see Figures 2, 6). For real world data, NATT can accurately detect e.g. jumps in Longitude and Latitude for rental prices in Amsterdam (see Figure 3).

Visual interpretation of categorical features, however, is often not very informative. One-hot encoding can lead to an excessive amount of shape functions. Especially for large tabular datasets with a lot of categorical features. One-hot encoding would require GAMs to fit each category with a separate shape function (e.g., Radenovic et al., 2022; Agarwal et al., 2021; Enouen and Liu, 2022). That leads to an increased amount of shape functions, that firstly need to be modeled and secondly need to be analyzed for interpretation. Label encoding can lead to ordinal interpretations in categories where no natural ordering is present thus impeding the intelligibility of the model. Target encoding can lead to problematic interpretations, especially for categories with very few observations, as the encoding for these categories may be unreliable or unstable.

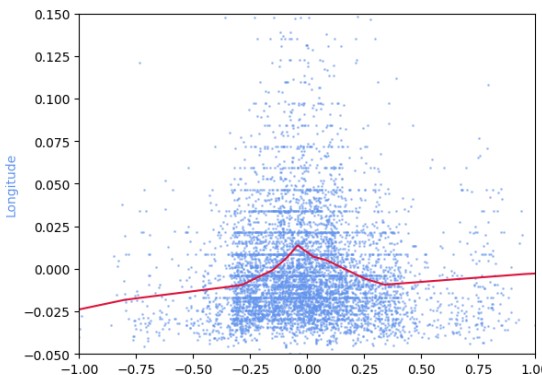

Figure 2: Shape function learned by NATT to predict rental prices in Amsterdam. The rental prices increase near the city center, depicted in the longitude graph.

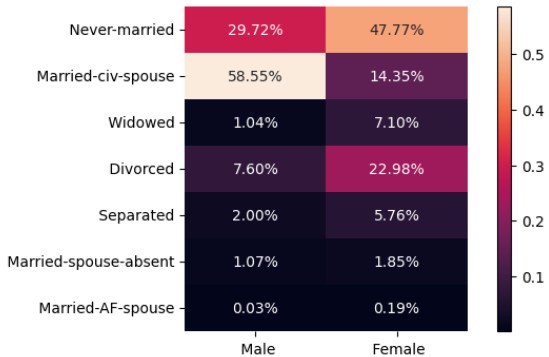

Figure 3: NATT Attention scores for the features *Marital Status* and *Gender* for the Adult dataset. A notable disparity exists between genders in the importance of the categorical attribute "Never-married" for income prediction in the model.

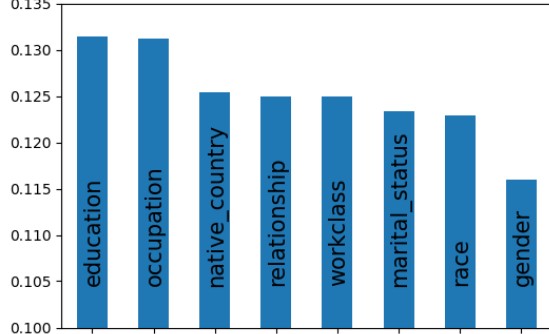

Figure 4: Categorical feature importances for the Adult dataset retrieved from the attention layers. The attention weights for the [CLS] token are extracted as detailed in Formula 5. See the supplemental material for details.

NATT can leverage multiple interpretable transformer structures to account for interpretable categorical features and remain fully intelligible across continuous features. Using [CLS] tokens, NATT can use the attention maps to weight the average attention scores of the [CLS] token with the overall importance of the encoded categorical features. The importance scores for the continuous features, as well as for all the combined categorical features, can be obtained due to the additivity constraint following a similar method as described in Agarwal et al. (2021).

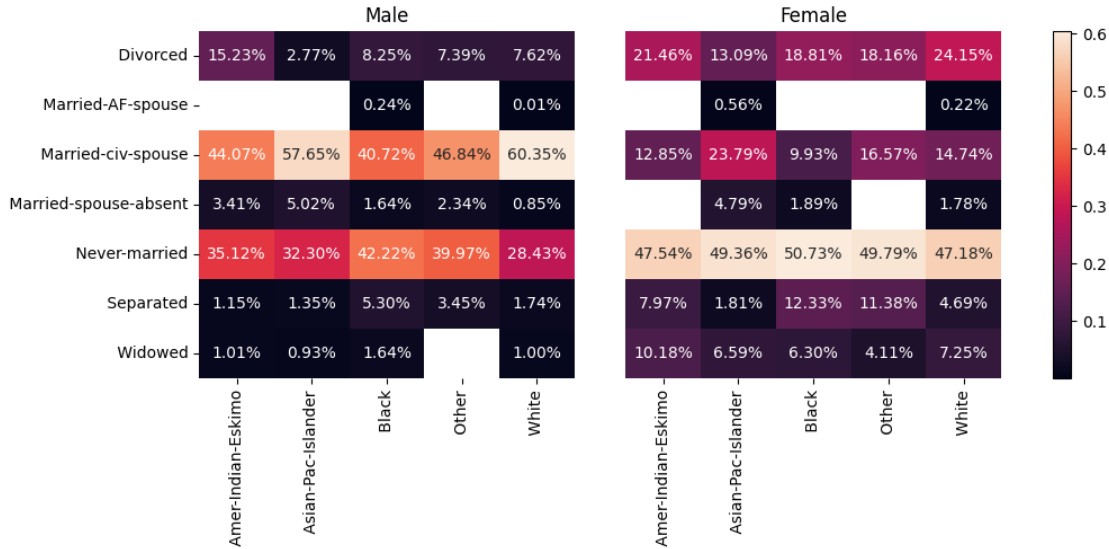

Figure 5: NATT attention scores for the features *Marital Status*, *Race* and *Gender* for the Adult dataset. The attributes "white" and "divorced" are significantly more important to the model's prediction of a person's income for females compared to males.

Thus, the importance for the categorical features over $n$ samples and $L$ attention heads is given by:

$$ I = \omega_{(cat)} \frac{1}{n} \sum_{i=1}^{n} \left( \frac{1}{H_{heads} \times L} \sum_{h,l} p_{i,h,l} \right), $$

where $\omega_{(cat)} \in [0,1]$ is the importance of all categorical variables, $h$ is the running index of the attention heads, $l$ denotes the $l-$th layer and $i$ denotes the $i-$th sample. $p_{i,h,l}$ thus denotes the $h$-th attention map for the [CLS] token from the forward pass of the $l$-th layer for sample $i$. Note, that since we are dealing with tabular data, the index or token position of a variable is fixed over all samples. Hence the models attention to a specific index of the input is equal to the models attention to that variable. Averaging the attention for every index over all samples and weighting with the overall categorical importance thus returns the average importance scores for every category.

Additionally, feature interactions between the categorical variables are always implicitly modeled. Attention scores for all possible categorical combinations can be extracted by summing not over all samples in Equation (5) but by summing only over the samples where the interaction of interest appears. Pairwise categorical feature interactions can thus easily be visualized (see Figure 3). Even higher order feature interactions can be subtracted by filtering the corresponding samples respectively (see Figure 5).

It is important to note, that attention weights are not to be equated with importance (Jain and Wallace, 2019). However, Brunner et al. (2019) found attention weights to be meaningful and identifiable for shorter sequences. Since the present approach handles tabular data, the sequences are by nature most often comparably short and more specifically shorter than the attention head dimension. While Jain and Wallace (2019) questions the information from attention weights altogether, Wiegreffe and Pinter (2019) provides evidence that challenges the blanket dismissal of attention weights and demonstrates their utility in several contexts. Furthermore,

the work by Gorishniy et al. (2022) is particularly relevant to our approach, since it leverages attention weights in a tabular setting. By comparing tabular attention scores with established methods like Integrated Gradients (Sundararajan et al., 2017), Gorishniy et al. (2022) finds consistently high correlation between attention maps and permutation test's feature importances. Our conducted simulation study (section 4.1) additionally showcases the robustness of the attention weights (Figure 6).

## 4 Experiments

### 4.1 Simulation study

For a small simulation study, we simulate 5000 observations with 2 continuous variables and 4 categorical variables (see Supplemental Material 7.5 for details). We train 100 models for 100 epochs each and visualize the learned shape functions for the continuous features in Figure 2a-2b. Both data generating functions are adequately captured by NATT.

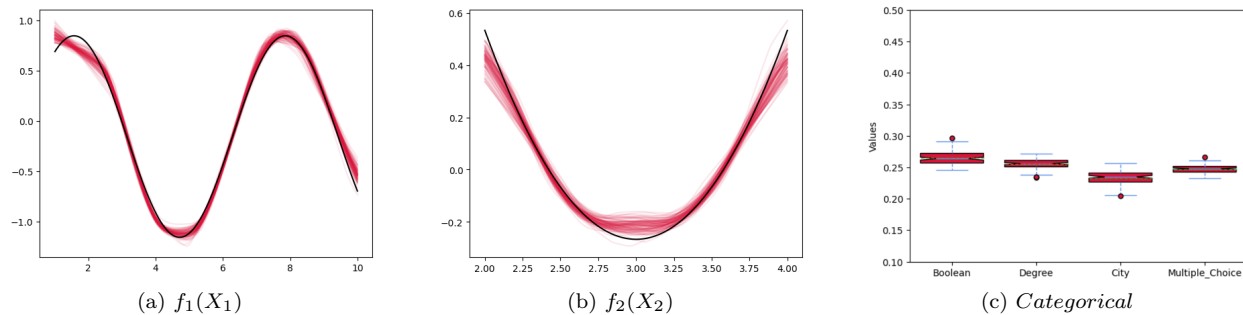

(a) $f_1(X_1)$       (b) $f_2(X_2)$       (c) *Categorical*

Figure 6: Shape functions and categorical feature importance learned by NATT for the simulation study. 100 models are fit and each trained model is visualized in red. The data generating function is visualized in black.

To test whether continuous feature interactions are adequately recognized, we use the same data generating process but add $f_1(X_1) \times f_2(X_2)$ to $y$. We visualize a random draw from multiple model fits and find that the complicated interaction patterns are adequately disentangled by NATT.

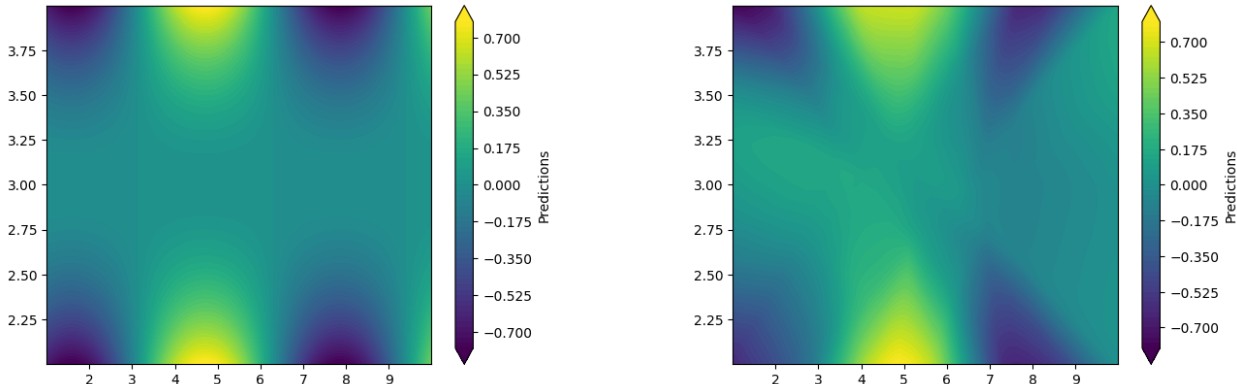

Figure 7: True feature interaction effect       Figure 8: NATT predicted feature interaction effect.

For categorical feature importance, our simulation consistently yields robust estimations (see Figure 2c). We find an average rank correlation of 0.76 between the true importance and the estimated importance scores over 100 fits which is comparable to XGboost with 0.79. Additionally, we compute the Kendall Tau score for demonstrating that NATT accurately detects the correct ordering of feature importance, compared to the

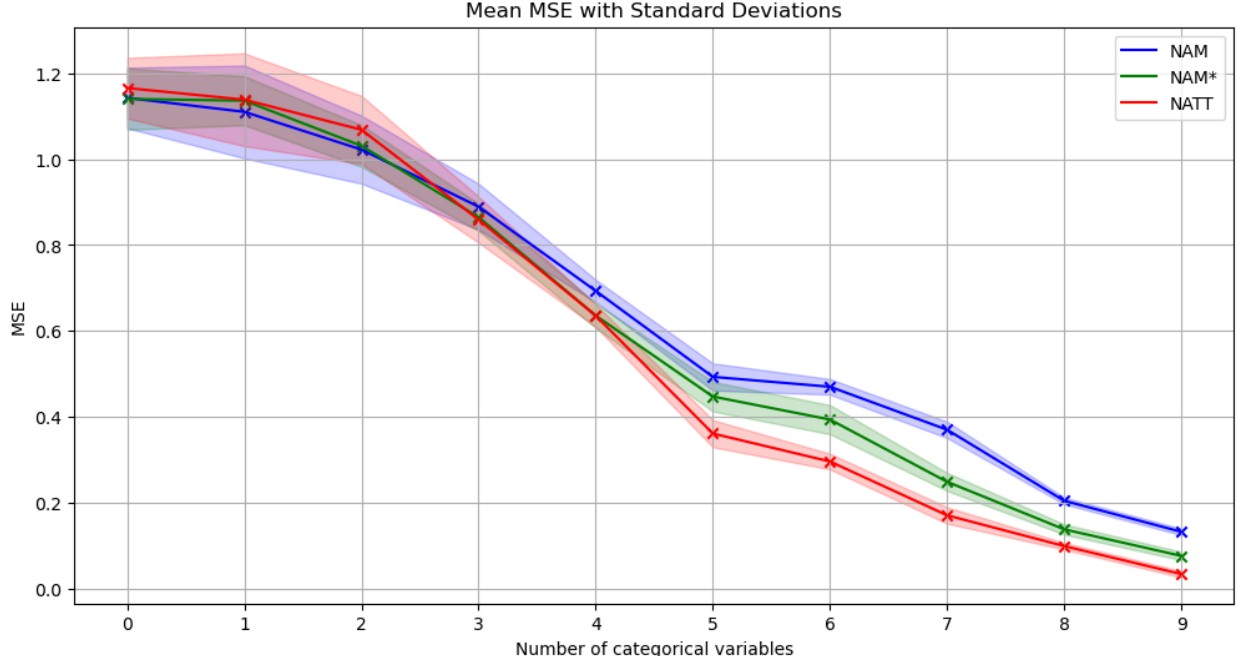

Figure 9: Performance of NATT, NAM and $\text{NAM}^{M*}$ that jointly models all categorical variables in a single MLP. With an increasing number of categorical features, NATT starts to outperform both NAM architectures.

overall effect the categorical features have on the dependent variable. We find a Kendall Tau score of 0.6 (compared to 0.0 for random ordering) and an accuracy of 65% for the ordering.

### 4.1.1 Ablation study

To further analyze NATT's capabilities, we conduct a small ablation study. We simulate a simple regression task dataset with 10 categorical variables and 3 numerical variables. The target variable is dependent on all variables through single effects as well as randomized interaction effects. For comparative analysis, we perform 5-fold cross-validation and train a NAM, NATT, and a NAM model that jointly models all categorical features in a single MLP. Note that this architecture loses the interpretability of categorical features altogether. For network architectures, we orient ourselves on Radenovic et al. (2022) and use single feature nets with $[64, 32, 32]$ neurons, ReLU activation, and 0.1 dropout after each layer. We use the same architecture for all models and employ an embedding size of 64 for NATT as well as 4 transformer blocks. We start with a learning rate of $1^{-03}$ and implement learning rate decay with a patience of 15 epochs and early stopping after 25 epochs of no improvement in the validation loss. All results are achieved with the models' best-performing weights on the validation dataset.

Since the NAM and NATT architectures are identical when there are no categorical features available, $g(\mathbb{E}\,[y]) = \beta_0 + \sum_{j=1}^{J} f_j(x_{j(cont)})$, we analyze the effect of the number of categorical features modeled. Figure 9 shows that the more categorical features are available, the better NATT performs compared to classical NAM structures. Similar to Gorishniy et al. (2022), we find that the Transformer architecture can increase performance in tabular problems. Note that for fewer categorical features, all models perform within their standard deviation bands as expected due to their similarity in architecture.

### 4.2 Datasets

We validate the effectiveness of our model on 8 machine learning benchmark datasets for both classification and regression. We perform 5-fold cross validation on all datasets and report the average performance as well as the standard deviations. For the classification tasks we report the Area under the curve (AUC). For

the regression tasks we report the root mean squared error (RMSE). Preprocessing is performed as done by Agarwal et al. (2021). We use the Vanilla implementation of NATT as described in Section 3 and thus even leave room for improvement by adapting more refined architectures (Gorishniy et al., 2021; Arik and Pfister, 2021). See the supplemental material for improved performances when leveraging [CLS] tokens during prediction.

**Classification datasets.** We report performance on the **Adult** dataset for predicting a persons income (Kohavi et al., 1996), the **Titanic** dataset retrieved from Kaggle, for predicting the survival of titanic passengers, the **Churn** dataset retrieved from Kaggle, covering whether a customer left a bank or not, and the **Insurance** dataset.

**Regression datasets.** We report performances on another **Insurance** dataset (Lantz, 2019), 2 **AirBnB** datasets with data from the cities of Munich and Amsterdam. Lastly we include the **Abalone** dataset retrieved from the UCI (Dua and Graff, 2017) as a dataset with only a single categorical variable and 3 categories. See the supplemental material for information about the datasets.

### 4.3 Baselines

We train the following state-of-the-art interpretable models as well as state-of-the-art *black-box* models: **Gradient Boosted Trees (XGBoost)**: Popular decision tree based gradient boosting, often outperforming DNNs on tabular data. We use the implementation provided by Chen and Guestrin (2016).
**Deep Neural Network (DNN)**: Unrestricted fully connected deep neural network trained with either a mean squared error loss function (regression) or cross entropy (classification).
**Tabular Transformer Networks (TabTransformer)**: A tabular transformer network followed by a fully connected MLP as introduced by Huang et al. (2020).
**Neural Additive Models (NAMs):** Linear combination of DNNs as described in Equation (1) and presented by Agarwal et al. (2021). Note that NA$^2$Ms do not scale to account for all feature interactions (Dubey et al., 2022). Pairwise interactions are thus only implemented between the numerical features.
**Explainable Boosting Machines (EBMs)**: State-of-the-art Generalized Additive Models leveraging shallow boosted trees (Nori et al., 2019).

Table 1: Comparison between NATT and a NAM for 4 classification and 4 regression datasets. The evaluation metrics are AUC and RMSE. We express the performance improvement of the NATT over the NAM as a percentage. The results are based on 5-fold cross-validation.

| Dataset Classification | NAM | NATT | Gain (%) |
|---|---|---|---|
| AUC ↑ | | | |
| Titanic | 84.9 | 86.5 | **1.9** |
| Adult | 91.0 | 91.4 | **0.4** |
| Insurance | 91.4 | 91.6 | **0.2** |
| Churn | 85.1 | 85.2 | **0.1** |
| Dataset Regression | Baseline NAM | NATT | Gain (%) |
| RMSE ↓ | | | |
| Insurance | 0.208 | 0.191 | **8.2** |
| Munich | 0.060 | 0.052 | **13.3** |
| Amsterdam | 0.042 | 0.037 | **11.9** |
| Abalone | 2.25 | 2.24 | **0.4** |
| Average Increase: | | | **4.9** |

**Neural Generalized Additive Model (NodeGAM)**: State-of-the-art Generalized Additive Models leveraging Neural Oblivious Decision Trees (Chang et al., 2021).

For neural network approaches, we take the same hyperparameters across all methods to provide a more consistent comparison and orientate on the benchmarks performed by Radenovic et al. (2022). For NodeGAM and EBM we use default values. All chosen hyperparameter settings allow to recover or even exceed performances reported in the literature (Chang et al., 2021; Huang et al., 2020; Thielmann et al., 2024).

### 4.4 Results

NATT outperforms NAMs on all datasets as shown in Table 1. We use the baseline model architecture for NATT as well as NAMs and do not account for feature interactions. Additionally, we demonstrate that NATT consistently outperform NAMs while remaining more parameter-sparse when using similar NAM architectures to Radenovic et al. (2022) and using much parameter sparser architectures than Agarwal et al. (2021). In our experiments, we decrease the amount of parameters in NATT in comparison to NAMs as defined by Radenovic et al. (2022); Dubey et al. (2022); Agarwal et al. (2021) by more than 40% (see Table 7). We use identical architectures for all numerical features for NAMs and NATT. The parameter difference can thus be attributed to the way the categorical features are modeled in the NAM.

Table 2: Average Rank table for interpretable models over all datasets. NATT and NA$^2$TT perform best considering other interpretable methods.

| Model | avg. Rank |
|---|---|
| **NATT** | **1.1** |
| NAM | 3.5 |
| NodeGAM | 2.4 |
| EBM | 3.0 |
| **NA$^2$TT** | **1.6** |
| NA$^2$M | 3.6 |
| NodeGA$^2$M | 2.6 |
| E$^2$BM | 2.1 |

Table 3: Comparison of 4 regression tasks for different interpretable as well as black-box models. The results are achieved with 5-fold cross validation. The average rmse values as well as standard deviations are reported. Best results per model type are marked in bold. The $^2$ denote pairwise feature interactions.

| | | RMSE ↓ | | | |
|---|---|---|---|---|---|
| Type Model | Insurance | Munich | Amsterdam | Abalone | |
| Blackbox | XGBoost | **0.165** (±0.008) | 0.053 (±0.016) | 0.039 (±0.014) | 2.30 (±0.05) |
| | DNN | 0.183 (±0.014) | 0.110 (±0.045) | 0.076 (±0.008) | **2.13** (±0.06) |
| | TabTransformer | 0.212 (±0.013) | **0.049** (±0.10) | **0.038** (±0.014) | 2.56 (±0.10) |
| Additive Models | NAM | 0.208 (±0.015) | 0.060 (±0.016) | 0.042 (±0.014) | 2.25 (±0.09) |
| | NodeGAM | 0.194 (±0.040) | 0.054 (±0.017) | 0.038 (±0.015) | 2.25 (±0.08) |
| | EBM | 0.194 (±0.004) | 0.053 (±0.017) | 0.041 (±0.014) | 2.27 (±0.06) |
| | NATT | **0.191** (±0.011) | **0.052** (±0.017) | **0.037** (±0.014) | **2.24** (±0.08) |
| Pairwise Feature Interactions | NA$^2$M | 0.205 (±0.016) | 0.056 (±0.015) | 0.042 (±0.013) | 2.11 (±0.05) |
| | NodeGA$^2$M | 0.194 (±0.040) | 0.051 (±0.017) | 0.037 (±0.015) | 2.13 (±0.05) |
| | E$^2$BM | **0.145** (±0.004) | 0.048 (±0.017) | 0.041 (±0.014) | 2.22 (±0.05) |
| | NA$^2$TT | 0.149 (±0.009) | **0.047** (±0.019) | **0.036** (±0.015) | **2.10** (±0.06) |

Tables 3 and 4 show the results of all models over all datasets, divided into three categories: Black-box models, interpretable additive models, and interpretable additive models accounting for pairwise feature interactions. The best model of each category is marked bold. Notably, the baseline NATT version performs comparably to a *black-box* XGBoost model and outperforms a fully connected Deep Neural Network on almost all datasets.

Moreover, NATT allows for better scaling when including feature interactions than NAMs. For instance, building feature interactions for all possible pairwise feature interactions on the Adult dataset when using one-hot encoding would result in more than 4500 shape functions in classical NAM approaches. Thus, fitting a NA$^2$M for such datasets would require an additional feature interaction selection step (Enouen and Liu, 2022). Even shared basis functions across all shape functions as proposed by Radenovic et al. (2022) may encounter recursion problems for larger datasets. Therefore, the results presented for NA$^2$Ms in this section only consider feature interactions between all numerical features.

Table 5: Number of Parameters for NAMs vs. NATT

| Model | Classification | | | |
|---|---|---|---|---|
| | Adult | Titanic | Insur. | Churn |
| NAM | 127K | 102K | 127K | 111K |
| NATT | 110K | 79K | 110K | 68K |

| Model | Regression | | | |
|---|---|---|---|---|
| | Insur. | Munich | Amst. | Abalone |
| NAM | 96K | 325K | 299K | 86K |
| NATT | 66K | 106K | 106K | 92K |

Table 4: Comparison of 4 classification tasks for different interpretable as well as black-box models. The results are achieved with 5-fold cross validation. The average AUC values as well as standard deviations are reported. Best results per model type are marked in bold. The $^2$ denote pairwise feature interactions.

| Type | Model | AUC ↑ | | | |
| | | Adult | Titanic | Insurance | Churn |
|------|-------|-------|---------|-----------|-------|
| Blackbox | XGBoost | **92.9** ($\pm$0.5) | 85.6 ($\pm$4.6) | **92.8** ($\pm$0.3) | **84.6** ($\pm$1.6) |
| | DNN | 90.6 ($\pm$0.5) | 84.1 ($\pm$8.0) | 90.5 ($\pm$0.4) | 83.7 ($\pm$1.3) |
| | TabTransformer | 91.0 ($\pm$0.5) | **86.1** ($\pm$4.0) | 91.1 ($\pm$0.4) | 84.6 ($\pm$1.2) |
| Additive Models | NAM | 91.1 ($\pm$0.3) | 84.9 ($\pm$4.3) | 91.4 ($\pm$0.4) | 85.1 ($\pm$1.1) |
| | NodeGAM | **91.5** ($\pm$0.4) | 82.8 ($\pm$8.6) | 91.6 ($\pm$0.5) | 85.1 ($\pm$1.4) |
| | EBM | 90.9 ($\pm$0.5) | 86.2 ($\pm$3.5) | 91.1 ($\pm$0.4) | 85.0 ($\pm$0.9) |
| | NATT | 91.4 ($\pm$0.3) | **86.5** ($\pm$3.7) | **91.6** ($\pm$0.3) | **85.2** ($\pm$1.4) |
| Pairwise Feature Interactions | NA$^2$M | 91.4 ($\pm$0.3) | 85.8 ($\pm$4.0) | 91.4 ($\pm$0.5) | 85.9 ($\pm$1.4) |
| | NodeGA$^2$M | 91.6 ($\pm$0.3) | 84.9 ($\pm$2.0) | 91.7 ($\pm$0.4) | 86.6 ($\pm$1.5) |
| | E$^2$BM | **91.9** ($\pm$0.3) | 86.8 ($\pm$4.6) | **92.0** ($\pm$0.4) | 86.1 ($\pm$1.4) |
| | NA$^2$TT | 91.5 ($\pm$0.3) | **87.0** ($\pm$4.0) | 91.6 ($\pm$0.3) | **86.9** ($\pm$1.6) |

Overall, we can validate the findings from Huang et al. (2020) and Gorishniy et al. (2022): Incorporating transformer embeddings when analyzing tabular data can improve the overall model prediction. However, we still find that XGBoost slightly outperforms TabTransformers by a slight margin of 5:3 on the benchmark datasets. Among all interpretable models, i.e., NATTs, NAMs, NodeGAMs and EBMs, NATT performs best in its base version achieving the lowest overall average rank over all datasets (see Table 2). When accounting for feature interactions, our method (NA$^2$TT) and boosting E$^2$BM improve in performance, while others benefit comparably less.

## 5 Conclusion

In this paper, we present NATT, a novel model architecture that offers interpretability. Our experiments demonstrate that NATT outperforms other neural interpretable methods across various datasets. While the additivity constraint ensures easy interpretation, this interpretation comes at a price. Throughout all interpretable models, we experience a performance trade-off in terms of predictive power. Full *black-box* models are still more performant than their glass-box counterparts. However, we demonstrate that NATT is a further step in the direction of closing that gap. Additionally, we find that when accounting for feature interactions, NA$^2$TT as well as E$^2$BM and NodeGA$^2$M are on par with *black-box* models. Throughout our benchmarks, we find that NATT only performs around 0.4% worse than the best *black-box* benchmark models. It is important to note that NATT's architecture allows further adaptations especially in the realms of modeling interactions. One potential approach is to leverage the trained transformer embeddings and train pairwise tensor-product feature interactions on the residuals using the second-to-last output layer of the shape functions. This strategy has the potential to greatly improve the performance of NATT, as it would allow for the joint learning and pre-training of the embeddings, leveraging the findings from Gorishniy et al. (2022).

## 6 Limitations and Future Work

A primary limitation of the current work is its focus on tabular data. Extending the presented approach to account for high dimensional input data such as documents or images is a key direction for further integrating interpretability into the domain of Deep Learning. By leveraging the transformer architecture for tabular features, NATT demonstrates the possibilities of integrating different network structures and shape functions into a single modeling framework. Through multiple experiments the generalizability of the structure is demonstrated. The interpretability of NATT, while being much more interpretable than *black-box* models, still lacks the inherent statistical inference of classical GAMs. Adaptations to include significance statistics

could be a further step towards Deep Learning models being deployed in high risk domains. Accounting for the underlying data distributions can already be accounted for, by extending NATT to account for all distributional parameters, as similarly done by Thielmann et al. (2024) (see Supplemental Material 7.1).

**Acknowledgements**

Funding by the Deutsche Forschungsgemeinschaft (DFG, German Research Foundation) within project 450330162 is gratefully acknowledged.

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

## 7 Supplemental Material for NATT

### 7.1 Beyond the mean: Location Scale and Shape

NATT is easily adaptable to account for arbitrarily many distributional parameters. Given a distribution with $K$ distributional parameters, a small architecture adaptation lets NATT become **N**eural **A**dditive **T**abular **T**ransformer Networks for **L**ocation **S**cale and **S**hape (NATTLSS). Similar to NAMLSS (Thielmann et al., 2024), we let each shape function have a K-dimensional output, with output $k = 1, 2, \ldots, K$ accounting for distributional parameter $k$.

The loss function gets adapted from e.g. the mean squared error, to the corresponding negative log-likelihood, which is dependent on the distributional parameters. Hence, NATTLSS minimizes $-\log(\mathcal{L}(\theta|y))$. Possible parameter interactions between the k-parameters are accounted for in the feature dependent shape-functions. Each distributional parameter is hence modeled by:

$$h(\theta^{(k)}) = \beta_0^{(k)} + \sum_j^J f_j^{(k)}(x_{j(cont)}) + f_{(cat)}^{(k)}(H^{(k)}(\boldsymbol{E}_\phi(\boldsymbol{x}_{cat}))), \tag{5}$$

where the superscript $^{(k)}$ denotes the $k-$th distributional parameter. $h(\cdot)$ thus denotes a parameter specific output layer activation or link function. E.g. a Softplus activation to account for the variance parameter in a normal distribution as the variance must be positive. $\beta_0^{(k)}$ denotes the parameter specific intercept. $f_j^{(k)} : \mathbb{R} \to \mathbb{R}$ denote the parameter-feature shape functions for the continuous features and $f_{(cat)}^{(k)} : \mathbb{R}^{(d \times j + c + 1)} \to \mathbb{R}$ denotes the parameter-feature shape functions for the categorical features. $H^{(k)}$ denotes the transformer layers, which are distributional parameter-specific. Leveraging pre-trained transformer networks, could allow to use a global $H(\cdot)$ over all parameters, however, jointly learning the feature embeddings for each distributional parameter is a more straight-forward architecture. Figure 10 demonstrates the advantages that NATTLSS can have over a simple NAM approach. While the overall predictive power might be similar, NATTLSS is much more faithful to the underlying data distribution and also accounts for the aleatoric uncertainty in the data.

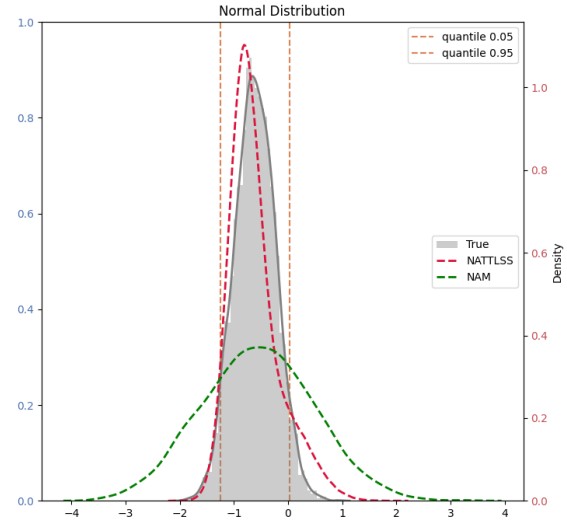

Figure 10: NATTLSS prediction vs. the prediction of a simple MLP for a normally distributed response variable.

Similar to Thielmann et al. (2024) we could adapt the network architecture, such that each shape function has a k-dimensional output layer, $f_{(cat)}^{(k)} : \mathbb{R}^{(d \times j + c + 1)} \to \mathbb{R}^k$ and $f_j^{(k)} : \mathbb{R} \to \mathbb{R}^k$. The overall model output would thus also be $k$-dimensional, with the $k$-th dimension accounting for the $k$-th distributional parameter. Thus, the adaptation to location scale and shape can be achieved with the same amount of shape functions as the mere mean prediction.

### 7.2 Datasets

We evaluated NATT's performance on multiple datasets, all having varying characteristics. NATT also perform well on smaller datasets, a property which could be furthermore increased by leveraging pre-trained networks for the transformer parts and fine-tuning the shape function, $f_{(cat)}$. For datasets with fewer categorical features, e.g. Abalone and Churn, the performance differences between NAM and NATT are not as large as for datasets with more categories (Munich, Amsterdam). Surprisingly, the differences for the

Adult dataset with the most categorical features are also comparably small, which can be attributed to the overall smaller effect the categorical variables have on the target variable.

Table 6: Statistics of the benchmark datasets.

| Dataset | No. Samples | No. Features | No. Categorical | No. Categories | Task |
|---------|-------------|--------------|-----------------|----------------|------|
| Insurance | 1338 | 6 | 3 | 8 | Regression |
| Abalone | 4177 | 8 | 1 | 3 | Regression |
| Munich | 4568 | 10 | 2 | 29 | Regression |
| Amsterdam | 6998 | 10 | 2 | 26 | Regression |
| Adult | 48842 | 13 | 8 | 102 | Classification |
| Churn | 10000 | 10 | 2 | 5 | Classification |
| Titanic | 627 | 10 | 9 | 19 | Classification |
| Insurance | 32561 | 13 | 8 | 102 | Classification |

### 7.2.1 Preprocessing

We implement the same preprocessing for all used datasets and only adjust it to specifically optimize it for different model architectures. We standard normalize the target variables for the regression problems. We closely follow Gorishniy et al. (2021) in their preprocessing steps and use the preprocessing also implemented by Agarwal et al. (2021). All numerical variables are scaled between -1 and 1. In contrast to Gorishniy et al. (2021) we do not implement quantile smoothing, as one of the biggest advantages of neural models is the capability to model jagged shape functions. Where needed, the categorical features are one-hot encoded (e.g. not for TabTransformer or NATT). We use 5-fold cross-validation and report mean results as well as the standard deviations over the folds.

### 7.3 NATT: Further Results

The reported benchmarks in section 4.3 for NATT are all performed with the Vanilla NATT architecture. Leaving the overall model structure intact, appending the [CLS] tokens to the column embedding and subsequently only feed the [CLS] tokens into the shape function $f_{(cat)}$ leads to very similar, and on average even a little bit better results. Therefore, by obtaining interpretable attention maps from the categorical features, we can further enhance performance.

Table 7: NATT performances when appending [CLS] tokens to the column embeddings and using the [CLS] tokens as inputs for the shape function $f_{(cat)}$.

| | Classification | | | |
|------|-------|---------|-------|-------|
| | Adult | Titanic | Insur. | Churn |
| NATT | 91.4 ($\pm0.3$) | 87.3 ($\pm3.5$) | 91.6 ($\pm0.3$) | 85.1 ($\pm1.2$) |
| | Regression | | | |
| | Insur. | Munich | Amst. | Abalone |
| NATT | 0.191 ($\pm0.011$) | 0.048 ($\pm0.017$) | 0.037 ($\pm0.013$) | 2.21 ($\pm0.07$) |

### 7.4 Hyperparameters

As outlined in section 4.3, we adopted consistent hyperparameters across all methods to ensure a fair comparison, drawing upon the benchmarks established by Radenovic et al. (2022) and Dubey et al. (2022) for our settings. For NodeGAM, EBM and XGBoost, we relied on default values, only fixing the number of interaction terms for EBM in the respective model class (EBM vs. EB$^2$M). For NAMs, we followed Radenovic et al. (2022). Importantly, our chosen hyperparameters allow us to match or surpass performance metrics reported in prior literature (Chang et al., 2021; Huang et al., 2020; Thielmann et al., 2024) that involved

extensive hyperparameter optimization. Additionally, we echo the sentiments of Bouthillier et al. (2021) regarding the complexities of comparing models with tuned hyperparameters.

Further supporting our approach, Grinsztajn et al. (2022) conducted extensive hyperparameter tuning across various tree-based and deep learning tabular models. Their findings indicated that while tuning generally benefits Transformer models more, the variability (large standard deviations) in performance across different hyperparameter sets was significant. Crucially, their work suggests that hyperparameter tuning does not consistently alter the relative performance rankings of models compared to evaluations conducted without tuning. This insight aligns with our focus on interpretability and the practicality of models for everyday users, where simpler, less parameter-intensive models are preferable to avoid the risks of overfitting inherent in tabular datasets with limited sizes.

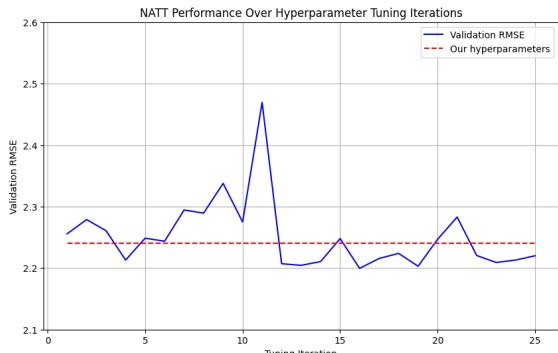

Figure 11: Optuna Hyperparameter tuning iterations for the NATT architecture for the Abalone dataset. The red line indicates our chosen hyperparameter setting. With more exhaustive hyperparameter tuning for our benchmarks, the performance for NATT, due to the Transformer architectures could be further improved. Note, that the large deviations on the left all stem from larger architectures that tend to overfit for limited tabular datasets.

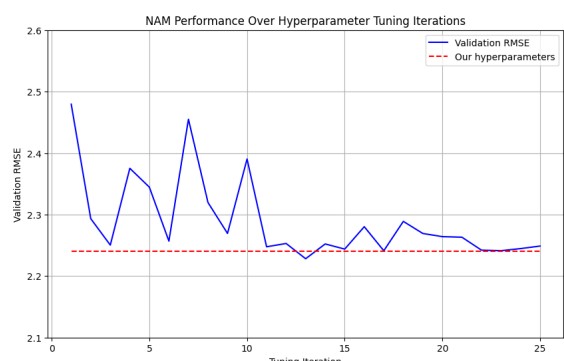

Figure 12: Optuna Hyperparameter tuning iterations for the NAM architecture for the Abalone dataset. The red line indicates our chosen hyperparameter setting. our findings support the results from Radenovic et al. (2022). Simple architectures in the NAM architecture are often sufficient and perform on par with larger architecture. Note, that the large deviations on the left all stem from larger architectures that tend to overfit for limited tabular datasets.

### 7.5 Data generating process for simulation study

In this data generation process, we aim to create a synthetic dataset with both categorical and continuous variables. The dataset consists of a target variable $y$, two continuous variables $X_1$ and $X_2$, and four categorical variables $Categorical_1$, $Categorical_2$, $Categorical_3$, and $Categorical_4$. We generate a dataset with a total of 5000 data points. The continuous features are simulated as:

$$f_1(X_1) = \frac{5\sin(X_1)}{5}$$
$$f_2(X_1) = -\frac{2(X_2 - 3)^2}{5}$$

Categorical features are simulated by assigning a categorical expression to each sample. Each expression corresponds to a specific numerical value. Consequently, we can assess the importance of each categorical feature based on its average impact on the dependent variable, $y$. A higher numerical difference between categorical expressions, along with a greater numerical effect on $y$, indicates a higher overall feature importance.

$$Categorical_1 = \begin{cases} 1.5 & \text{if } = A \\ -1.5 & \text{if } = B \\ 0.0 & \text{if } = C \end{cases}$$

$$Categorical_2 = \begin{cases} 0.0 & \text{if } = Yes \\ -0.75 & \text{if } = No \\ 0.75 & \text{if } = Maybe \end{cases}$$

$$Categorical_3 = \begin{cases} 0.0 & \text{if } = Miami \\ 0.2 & \text{if } = NewYork \\ 0.2 & \text{if } = Chicago \end{cases}$$

$$Categorical_4 = \begin{cases} 1.0 & \text{if } = Bachelors \\ 1.0 & \text{if } = Masters \\ 0.0 & \text{if } = PhD \end{cases}$$

Thus, we randomly draw 5000 samples for each categorical variable, draw $X_1$ and $X_2$ from uniform distributions between the values 1 and 10 and 2 and 4.

$$y = f_1(X_1) - f_2(X_2) + Categorical_1 + Categorical_2 + Categorical_3 + Categorical_4$$

## 8 Attention scores for tabular transformer networks

In a tabular Transformer model that incorporates a [CSL] token alongside $J$ variables, the *sequence length* is equal to the number of variables and due to the [CSL] token extends to $J + 1$. Consequently, for a single sequence (all variables), the attention weight matrix within a layer $l$ and for an attention head $h$ is of size $(J + 1) \times (J + 1)$. This matrix accounts for the attention weights across all variables and the [CSL] token. The attention weight directed towards a specific position $j$ in the input sequence, which could represent any of the $J$ variables or the [CSL] token, is denoted as $p_{h,l,j}$. Thus, the importance of this position for the model's decision-making is expressed as $A_{h,l,j}$. Note, that position $j$ in the input sequence of tabular data references to variable $j$ of $J$ variables. Given that we only include the [CSL] token as the input for the downstream tasks, the attention scores of the [CSL] aggregate all necessary information. We can thus quantify the collective attention it receives from all other tokens and hence variables. This is achieved by aggregating the attention weights directed towards the [CSL] token across various heads and layers. Hence, we can analyze $A_{h,l,1}$, when 1 is the index of the [CLS] token, for all input samples and thus all contexts (feature interactions). Note, that we omitted the last index $j$ in the main body of our paper and used $A_{i,h,l,1} = A_{i,h,l}$ for the $i-$th sample for better readability.

