# OpenReview forum: "Interpretable Additive Tabular Transformer Networks"
_TMLR — Accepted by TMLR_

### Review · Reviewer_nMEX · 2024-02-04

**Summary Of Contributions:**

This paper proposed Neural Additive Tabular Transformer that incorporates categorical feature embeddings into an additive model architecture. The advantage of this method is better interpretability. The authors also show better performance of this method compared with other baselines.

**Audience:**

Yes

**Broader Impact Concerns:**

No such concers.

**Claims And Evidence:**

Yes

**Requested Changes:**

I failed to find any discussion and illustration in Tables 4 and 5. Why do you divide the models into three parts?

**Strengths And Weaknesses:**

Strengths: The paper is easy to follow overall. The proposed method seems reasonable.

Weaknesses: Some parts of the paper are unclear. Section 3.1 seems not so interesting to me. For example, why do we want Figures 3 and 5 to be visualized? What can it be used for in your tasks?

---

> ### Author Response · Authors · 2024-03-28
> **Answer to Reviewer nMEX**
>
> Dear Reviewer,
> Thank you for your comments and questions, particularly regarding the clarity and interest of Section 3.1. Below, we adress your concerns in detail and outline which measures we have taken to improve our mansucript according to your suggestions.
>
> > Some parts of the paper are unclear. Section 3.1 seems not so interesting to me. For example, why do we want Figures 3 and 5 to be visualized? What can it be used for in your tasks?
>
>  We firstly want to emphasize that interpretability is a crucial aspect of analyzing tabular data, especially when it comes to understanding model decisions in real-world contexts.
> To address your concerns, we have revised parts of Section 3.1 and the introduction to better emphasize the importance of interpretability in our study. We believe that these revisions will make the objectives and contributions of our work more transparent.
>
> Regarding the visualization of Figures 3 and 5, we appreciate your query on their relevance to our tasks. These figures are designed to capture the importance of interactions between categorical variables in our model's predictions. For instance, Figure 3 highlights how the categorical expression "Never married" significantly influences the model's prediction outcomes, demonstrating a notable disparity in its importance between genders in the context of income prediction. Figure 5 shows that the attributes "white" and "divorced" are significantly more important to the model's prediction of a person's income for females compared to males.
> We hope that these clarifications and the revised sections address your concerns and illustrate the critical role of these visualizations in enhancing the interpretability and applicability of our findings.
>
> Thank you for your valuable feedback, which has undoubtedly strengthened our manuscript.
>
>
> > I failed to find any discussion and illustration in Tables 4 and 5. Why do you divide the models into three parts?
>
> We have divided the tables into three parts to emphasize the distinctions in the respective models. We have added a column to clarify as well as added better descriptions. In a nutshell, the first section includes fully black-box models, the second part additive interpretable models, and the last section additive models that include pairwise feature interactions.

---

> > ### Comment · Reviewer_nMEX · 2024-04-04
> >
> > Thank you for addressing my concerns. I have no further questions.

---

### Review · Reviewer_B5UM · 2024-03-15

**Summary Of Contributions:**

This paper presents a Neural additive transformer as an 'interpretable' alternative to transformer models for tabular datasets. This line of work is inspired by generalized additive models, which restrict the shape of the function that is learned per input feature. The key conceptual challenge that this paper tackles is that, in normal GAMs or other Neural additive models, you need a different shape function per feature or feature interaction. However, when one has a categorical variable, the number of features increases and using a single function to represent each category is prohibitive. To sidestep this issue, this paper proposes to use a transformer to embed these categorical embeds. A single shape function is then learned for each group of features. The paper presents several empirical findings that show that the proposed NAT architecture matches the performance of other additive models and even Xgboost on several datasets.

**Audience:**

Yes

**Broader Impact Concerns:**

None.

**Claims And Evidence:**

Yes

**Requested Changes:**

Please see the weaknesses section above. None of the requested changes are disqualifying, but mostly require additional clarifications.

**Strengths And Weaknesses:**

Overall, this paper is well written and I enjoyed reading it.

## Strengths
- **Clear Exposition and Writing**: I found the problem statement clear and well articulated in this work. Each section of the paper also flows from one to another. Figure 1 is quite helpful and clearly demonstrates the problem that this work is trying to solve.
- **Comprehensive Experiments**: The experiments in this work clearly backup the claim made in the paper. The authors explore several datasets and clearly test across both previous additive models and black-box approaches too.
- **Simple Solution**: The solution proposed here is simple and easy to implement, which means that it can immediately be used in practice without too much hassle.
- It is quite nice that in some cases, the number of params for the NATT is smaller than the NAM, yet without substantial performance degradation.

## Weaknesses
- **Conceptual Clarity**: One challenge I have with the writing is that from the introduction, it still a bit unclear why it is a big deal to deal with categorical variable separately. Perhaps this point could be emphasized more in the current introduction.
- **Attention Weights Identifiability**: There is a large literature on identifiability of attention weights, see: "Attention is not explanation", "Attention is not not explanations", and "On identifiability of transformers". Since you need the attention weights in your feature importance estimate, how do we know that it is a reliable estimate?
- **Section 3.1**: Why is 'FIGURE LABEL' in red?
- Can you justify the feature importance formula somewhere in the appendix? I am missing something simple here I think. Why sum over n samples? Isn't the feature shape a 'global' quantity? Perhaps I am misunderstanding what this formula is computing. Is it the feature importance of a single categorical variable?
- This is a stylistic one: I think tables 4 and 5 should come before 2 and 3. Tables 2 and 3 are more about specific comparisons among interpretable architectures. While Table 4 and 5 are general.

---

> ### Author Response · Authors · 2024-03-28
> **Answer to Reviewer B5UM**
>
> Dear reviewer, thank you for your careful review. We have made adaptations to our mansucript according to your comments/suggestions, believe that it substantially improved our mansucript and want to thank you for that.
>
> Below we have adressed your concerns and outlined where we adapted our manuscript:
>
> > One challenge I have with the writing is that from the introduction, it still a bit unclear why it is a big deal to deal with categorical variable separately. Perhaps this point could be emphasized more in the current introduction.
>
>
> Thank you for your constructive feedback. We appreciate your insights on enhancing the clarity regarding the significance of addressing categorical variables separately in our work. In response to your concern, we have revised our introduction to better emphasize this aspect. We would also like to highlight that an important benefit of NATT, which stems from its approach to handling categorical variables separately, is a performance increase. Consequently, NATT narrows the gap between inherently interpretable models and their black-box counterparts.
>
> To further address your comment, we have added a new paragraph specifically aimed at clarifying the objectives of NATT.
>
> > There is a large literature on identifiability of attention weights, see: "Attention is not explanation", "Attention is not not explanations", and "On identifiability of transformers". Since you need the attention weights in your feature importance estimate, how do we know that it is a reliable estimate?
>
> Thank you for the critical literature on the identifiability of attention weights. We appreciate the opportunity to clarify and support the methodology used in our work.
>
> As discussed by [1] attention weights remain meaningful for shorter sequences, a condition met by the tabular data in our study. This distinction is crucial as it underscores the applicability of attention weights in our context, where the sequence lengths are well within the threshold where attention mechanisms maintain their interpretability.
> The debate between [2] and [3] enriches our understanding of the context-dependent nature of attention weights. The latter paper, in particular, provides evidence that challenges the blanket dismissal of attention weights as unreliable, demonstrating their utility in several contexts. This is in line with our findings and supports our use of attention mechanisms for feature importance estimation in tabular data.
> Furthermore, the work by [4] is particularly relevant to our approach. By comparing tabular attention scores with established methods like Integrated Gradients [5] and permutation test’s feature importances, they found a consistently high correlation, further validating the reliability of attention scores in this setting. Their results, which show a higher correlation than that achieved by Integrated Gradients, support our decision in the use of attention weights for feature importance estimation in tabular data.
> We believe these points address the concerns raised and underscore the reliability and applicability of attention weights in our study's context. Thank you for allowing us to clarify these aspects, and we hope this response has adequately addressed your concerns.
>
> To improve our manuscript accordingly, we have added a paragraph outlining our arguments above in section 3.1.
>
> > Can you justify the feature importance formula somewhere in the appendix? I am missing something simple here I think. Why sum over n samples? Isn't the feature shape a 'global' quantity? Perhaps I am misunderstanding what this formula is computing. Is it the feature importance of a single categorical variable?
>
> Thank you for this comment. We have included a short explanation in the main body of our work as well as a dedicated section in the appendix. In a nutshell, we are computing an average attention map over all samples for all categorical variables. Referencing Transformers in NLP, for our framework each variable is at a fixed index in the input. Hence, the average attention scores over all samples and thus all contexts/interaction effects for index $i$, are equal to the average attention scores for the single feature $i$.
>
>
> > Minor Comments
> > - Section 3.1: Why is 'FIGURE LABEL' in red?
> > - This is a stylistic one: I think tables 4 and 5 should come before 2 and 3. Tables 2 and 3 are more about specific comparisons among interpretable architectures. While Table 4 and 5 are general.
>
> Thank you for noticing. We have adapted this according to your suggestion.
>
>
> [1] Brunner, et al. On identifiability in transformers. 2019
>
> [2] Jain and Wallace. Attention is not explanation. 2019
>
> [3] Wiegreffe and Pinter. Attention is not not explanation. 2019
>
> [4] Sundararajan, et al. Axiomatic attribution for deep networks. ICML 2017
>
> [5] Gorishniy, et al. On embeddings for numerical features in tabular deep learning. NeurIPS 2022

---

> > ### Comment · Reviewer_B5UM · 2024-05-08
> > **Addresses my concerns**
> >
> > I would like to thanks the authors for the response and directly addressing the concerns I had.

---

### Review · Reviewer_osxC · 2024-03-22

**Summary Of Contributions:**

The paper proposes the Neural Additive Tabular Transformer Networks (NATT) which is a model that combines the strengths of additive neural networks for interpretability and Transformer models for handling categorical data. The method achieves high predictive performance.

NATT outperforms other interpretable deep learning models and achieves performance levels similar to complex "black box" models. This makes it a valuable tool in areas where explainability is critical alongside accurate predictions. The method can incorporates feature interactions (pairwise or higher-order), further boosting its performance and making it competitive even with non-interpretable models.
Experiments on diverse datasets demonstrate the superiority of NATT over interpretable baselines and its competitiveness with state-of-the-art tabular data models.

**Audience:**

Yes

**Broader Impact Concerns:**

I think the paper can be of importance to this particular machine learning community. I can not really judge this.
The experiments seem quite toyish, but I can not judge this since I do not know the related work.

**Claims And Evidence:**

No

**Requested Changes:**

Page 7, what is (see Figure 2c)?

Please provide all training details / architecture details.
Please report performance when changing hyperparameters / provide hyperparameter scan ranges for the method / baselines.
Please report  ablations: What happens if you exchange the Transformer with an e.g. MLP?

If you provide these analyses, I think the paper is fit for publication.

**Strengths And Weaknesses:**

The paper is clearly written and the method simple while outperforming NAMs and other baselines in all benchmarks.
Of course, generally speaking, the interpretability is limited since still black-boxes are incooportated in the architecture - but this is the same for NAMs.
For me, as a kind of outsider of this particular machine learning field, it would be nice to understand why the authors are not simply scale the models to obtain better numbers. Is it because you start overfitting? If this is the case, and actually generally, I would like to see a performance plot showing performance with changes to the number of layers / width / number of paramterers - I think this would especially make sense on the Transformer side, as this is the novel part of the model.
Also to fairly examine the baselines / the proposed method, can you provide numbers on sensibility the hyperparameters of the proposed method / baselines. I guess you scan hyperparemters for all methods?

---

> ### Author Response · Authors · 2024-03-28
> **Answer to Reviewer osxC**
>
> Dear reviewer, thank you for your careful review. We have made adaptations to our mansucript according to your comments/suggestions, believe that it substantially improved our mansucript and want to thank you for that.
>
> > Of course, generally speaking, the interpretability is limited since still black-boxes are incorportated in the architecture - but this is the same for NAMs.
>
> We acknowledge that while NATT, like all machine learning models incorporating black-box components, has inherent interpretability limitations, it's important to note that its interpretability is competitive with, if not superior to, that of similar ML/Deep Learning models such as NodeGAM, NAM, and EBM. Moreover, NATT distinguishes itself by delivering superior predictive performance in comparison to these counterparts.
>
>
> > For me, as a kind of outsider of this particular machine learning field, it would be nice to understand why the authors are not simply scale the models to obtain better numbers. Is it because you start overfitting? ...
>
> - We oriented our model architectures based on previously published studies to ensure a fair and consistent comparison framework. Specifically, we chose the architectures for NAMs from [1] and [2]. We used the default parameters for NodeGAM but outperformed their reported results on the same datasets [3].
> -  Additionally, for tabular problems characterized by limited dataset sizes and an emphasis on interpretability, increasing the model size often leads to significant overfitting. This overfitting not only hinders model performance on unseen data but also detracts from the model's interpretability - a key focus of our work. Therefore, in the context of interpretability and catering to everyday users, we prioritize less complex models with fewer parameters (See e.g. Table 1), as they provide a more balanced trade-off between predictive performance, memory- and time-efficient model fitting, as well as understandability.
> - We agree that performing extensive hyperparameter tuning could especially benefit the Transformer models. However, since NATT is the only interpretable model that leverages transformer layers, extensively tuning parameters would most likely benefit NATT more strongly than the other models. We have included a section on this in the supplemental material.
>
> [1] Dubey, et al. Scalable interpretability via polynomials. NeurIPS 2022
>
> [2] Radenovic, et al. Neural basis models for interpretability. NeurIPS 2022
>
> [3] Chang, et al. Node-gam: Neural generalized additive model for interpretable deep learning. ICLR 2021

---

> > ### Author Response · Authors · 2024-03-28
> > **Continued Answer to Reviewer osxC**
> >
> > > Also to fairly examine the baselines / the proposed method, can you provide numbers on sensibility the hyperparameters of the proposed method / baselines. I guess you scan hyperparemters for all methods?
> >
> > - As outlined in section 4.3, we adopted consistent hyperparameters across all methods to ensure a fair comparison, drawing upon the benchmarks established by [2] and [1] for our settings. For NodeGAM and EBM, we relied on default values. Importantly, our chosen hyperparameters allow us to match or surpass performance metrics reported in prior literature [3, 4, 5] that involved extensive hyperparameter optimization. Additionally, we echo the sentiments of [6] regarding the complexities of comparing models with tuned hyperparameters.
> >
> > - Further supporting our approach, [7] conducted extensive hyperparameter tuning across various tree-based and deep learning tabular models. Their findings indicated that while tuning generally benefits Transformer models more, the variability (large standard deviations) in performance across different hyperparameter sets was significant. Crucially, their work suggests that hyperparameter tuning does not consistently alter the relative performance rankings of models compared to evaluations conducted without tuning. This insight aligns with our focus on interpretability and the practicality of models for everyday users, where simpler, less parameter-intensive models are preferable to avoid the risks of overfitting inherent in tabular datasets with limited sizes. Furthermore, the practical circumstances where our methodology is employed might not permit hyperparamter optimization because of the associated potentially vast computational requirements of an extensive search over hyperparamteres. Assessing and comparing our method's performance without specifically adapting to the intecracies of individual datasets is thus in line with our idea of introducting a method aimed to be robust, interpretable, easy to use and to have profound predictive performance.
> > - Furthermore, the practical circumstances where our methodology is employed might not permit hyperparamter optimization because of the associated potentially vast computational requirements of an extensive search over hyperparamteres. Assessing and comparing our method's performance without specifically adapting to the intecracies of individual datasets is thus in line with our idea of introducting a method aimed to be robust, interpretable, easy to use and to have profound predictive performance.
> >
> >
> >
> > ## Minor Comments
> > > Page 7, what is (see Figure 2c)?
> >
> > - Figure 2c demonstrates the stable performance of NATT across various iterations, using a simplified data example. Just as observed with the numerical variables in Figures 2a and 2b, the model consistently captures the true effects within the categorical variables.}
> >
> > > Please report ablations: What happens if you exchange the Transformer with an e.g. MLP?
> >
> > - Thank you for pointing that out. We have included a dedicated section with an additional ablation study to compare NATT, classical NAMs and NAMs that model all categorical variables jointly in a single MLP as per your suggestion.
> >
> > [1] Dubey, et al. Scalable interpretability via polynomials. NeurIPS 2022
> >
> > [2] Radenovic, et al. Neural basis models for interpretability. NeurIPS 2022
> >
> > [3] Chang, et al. Node-gam: Neural generalized additive model for interpretable deep learning. ICLR 2021
> >
> > [4] Huang, et al. Tabtransformer: Tabular data modeling using contextual embeddings. 2020
> >
> > [5] Thielmann, et al. Neural Additive Models for Location Scale and Shape: A Framework for Interpretable Neural Regression Beyond the Mean. 2023
> >
> > [6] Bouthillier, et al. Accounting for variance in machine learning benchmarks. MLSys 2021
> >
> > [7] Grinsztajn, et al. hy do tree-based models still outperform deep learning on typical tabular data? NeurIPS 2022

---

### Public Comment · ~Calvin_McCarter1 · 2024-05-26
**Reproducibility questions**

This seems like a potentially very useful method, as it addresses an important problem and offers promising results. But I have two concerns:
 - As far as I can tell, no code is provided. Am I missing something? I would think that provided software (at the very least, released upon acceptance, if not provided during review) is table stakes in 2024.
 - Why are there no comparisons on various tabular dataset-of-dataset benchmarks? Because this method has something to offer (interpretability) that other DL methods don't have, I don't expect it to provide SotA predictive performance. But it would be good to know how much performance one tends to trade off for interpretability. This is especially significant because no code is provided, as mentioned above. Before I invest time in implementing another method, I'd like to have at least a general sense of the tradeoffs involved.

---

### Decision · Action_Editor_Sgi1 · 2024-05-09

**Recommendation:** Accept as is

**Comment:**

The reviews were quite positive from the beginning of the process, and the authors have successfully addressed the comments of the reviewers. I do not see any barrier for acceptance.

**Audience:**

This is an interesting yet simple method that contributes to the literature on additive models.

**Claims And Evidence:**

All reviewers agreed that the paper provided compelling evidence on the performance of their proposed approach.